# Experimental study on water-heat-salt migration and deformation characteristics of subgrade filler during freeze-thaw cycles in Northwestern China

Hang Mu[1], Wenhui Zhao[1]*, Ruiqi Wang[1], Ke Zhang[2]

1 School of Civil Engineering, Lanzhou Jiaotong University, Lanzhou, China, 2 Gansu Railway Construction Investment Group Co., Ltd., Lanzhou, China

* zhaowh1989@163.com

## Abstract

The seasonal freeze-thaw action of saline soils in Northwestern China deteriorates embankment material properties, thereby adversely affecting train ride comfort and increasing railway maintenance expenditures. To explore the mechanism of sulfate erosion on subgrade fillers, soil column freeze-thaw cycling tests were conducted with the background of the arching disease in the transition sections of ballastless railway culverts in Northwestern China. The water-temperature-salt migration and deformation characteristics of subgrade filler during freeze-thaw cycles are investigated. The results show that the magnitudes of temperature variations decrease upon increasing the distance from the top. Additionally, under alternating hot and cold cycles, a temperature lag occures at the lower sensors. For the multi-layer soil column, the water content increases with the number of freeze-thaw cycles. The salt content in the multi-layer soil column gradually migrates upward upon increasing the number of freeze-thaw cycles, and the overall salt content continuously increases, reaching the top of the column by the 5th cycle. During both single-layer and multi-layer freeze-thaw cycles, the accumulated residual deformation increases with the number of cycles. Component analysis and microstructural examinations reveal the formation of ettringite and gypsum within the samples, which are related to water-heat-salt migration.

## 1. Introduction

Global soil salinization is occurring rapidly, and as China's transportation network continues to improve, an increasing number of embankment projects are being constructed in salinized soil areas [1]. In the seasonal saline soil region of northwestern China, the presence of sulfate saline soils at the surface and the cold temperatures in winter severely impact the service performance of embankment materials under the freeze-thaw cycles and the combined effects of sulfate corrosion, which macroscopically manifests as upward heaving deformation of the track structure(arching

**Data availability statement:** All relevant data are available at the following DOI: 10.6084/m9.figshare.30661367.

**Funding:** The authors gratefully acknowledge the support from the National Natural Science Foundation of China (52368065), and the National Natural Science Foundation of ChangSha(kq2402025).The funders had no role in study design, data collection and analysis, decision to publish, or preparation of the manuscript.

**Competing interests:** The authors have declared that no competing interests exist.

disease). Trains are highly sensitive to deformations of subgrade structures when operating at high speeds, and even minor arching on the subgrade may affect the smoothness of the railway and passenger comfort. In severe cases, it may compromise the safety of train operations or significantly increase the workload and maintenance costs for railway engineering departments. According to the literature [2–6], it is crucial to understand the heaving mechanism in the transition section of culverts, as well as the migration patterns of water-heat-salt and the swelling characteristics of sulfates in embankment materials.

Many scholars have investigated the migration of water and salt and salt expansion characteristics. Yang et al. [7] studied the migration trends of water and salt in subgrade cushion layers with different ratios of loess and sand and found that the maximum rise height of water and salt occurred when the sand content in the fill reached 30%. Zhou et al. [8] established a mathematical model for the four-field coupling of water, heat, salt, and force in unsaturated saline soil. They used this model to calculate variations in each of the four fields in the soil and validated the model through indoor experiments. Zhou et al. [9] established a multi-field coupling model that considered the effects of temperature gradients, wherein they investigated the influence of salt adsorption on the heat and mass transfer processes and deformation characteristics of unsaturated saline soil. Li et al. [10] conducted unidirectional freeze-thaw tests with different salt contents and found that different phase change pathways led to different migration patterns of water and salt, with a higher salt content inhibiting water migration. Wang et al. [11] found that the water and salt migration rate was fastest when the fine particle content was 10%, and significant salt migration occurred after increasing the fine particle content to 30%. Zhang et al. [12] studied the water and salt migration law in areas with highly saline soil based on actual engineering projects and revealed the capillary rise height of the three undisturbed soils. In terms of the salt expansion characteristics. Jiang et al. [13] conducted freeze-thaw cycle tests on silty sulfate saline soil in a closed system and proposed that temperature and salt content were the main factors causing the expansion deformation of the soil during freeze-thaw cycles. Chen et al. [14] investigated variations in the salt expansion deformation of coarse-grained soil after compaction upon increasing the number of freeze-thaw cycles. Their results showed that salt expansion occurred in the fill after a small number of freeze-thaw cycles. Wang et al. [15] found that the deformation amount of saline soil with different salt contents exhibited an M-shaped curve upon increasing the fine particle content. The deformation reached local maxima at fine particle contents of 15% and 10%. Li et al. [16] proposed a coupled mathematical model for water-heat-air-salt-force (HTASM) that considered the effects of water melting, sodium sulfate crystallization, and steam flow during the freezing process based on unsaturated soil mechanics and porous media theory. Zhang et al. [17] described the growth process of ice crystals and salt crystals using crystallization kinetics theory. They established a coupled mathematical model for water-heat-salt-force that considered the effects of phase changes in saturated frozen sulfate saline soil. They found that the pore pressure generated by phase changes was the main driving force for water and salt migration and the deformation of porous media.

In the setting of the frost heave disease occuring at transition section of a ballastless railway embankment in north-west China. The vertical migration patterns of water-heat-salt in subgrade fillers are analyzed through laboratory freeze-thaw cycles. The arching mechanism in saline soil on the subgrade transition section areas is revealed. The results are expected to provide engineering references for similar projects in the region.

## 2. Experimental material and scheme

### 2.1 Experimental materials

Based on actual engineering projects, three types of fillers were used: graded crushed stone, cobble and gravel stone, and fine round gravel soil. Sieve analysis and compaction tests were conducted following the method outlined in "Geo-technical Testing Code for Railway Engineering" (TB 10102−2023). The test results are shown in Table 1. Ordinary Port-land cement (P•O42.5) was used for all experiments.

### 2.2 Experimental scheme

Single-layer and multi-layer soil column freeze-thaw cycling tests were conducted. For the single-layer soil column freeze-thaw cycle (Fig 1(a)), the soil column test mold is 30 cm high and 30 cm in diameter. It is filled with 5% cement-stabilized graded crushed stone, 3% cement-stabilized graded crushed stone, 5% cement-stabilized cobble and gravel stone, and fine round gravel soil. The moisture content is controlled around 4.75%, 5.51%, 4.38%, and 3.07%. Layered compaction is carried out using a homemade circular tamper to achieve a compaction degree similar to those in the field, which are 0.97, 0.95, 0.95, and 0.92. Three sensors are buried at intervals of 10 cm from the bottom of the mold at a depth of 5 cm and are labeled No. 1 to No. 3 from the top to bottom. The multi-layer soil column freeze-thaw cycle test is illustrated in Fig 1(b), with a 4-layer soil column test mold consisting of the same materials as the single-layer test. The compaction, moisture

**Table 1. Basic physical parameters of soil samples.**

| Filler Material | Maximum Dry Density/(g·cm⁻³) | Optimal Moisture Content/% | Mass Percentage Below a Particle Size/% | | | | | |
|---|---|---|---|---|---|---|---|---|
| | | | 31.5 mm | 22.4 mm | 7.1 mm | 1.7 mm | 0.5 mm | 0.075 mm |
| **Graded Crushed Stone** | 2.42 | 3.71 | 97.58 | 82.68 | 47.41 | 26.39 | 18.00 | 8.12 |
| **Cobble and Gravel Stone** | 2.17 | 3.56 | 0 | 96.00 | 50.10 | 30.69 | 19.05 | 9.57 |
| **Fine Round Gravel Soil** | 2.27 | 3.07 | 0 | 0 | 97.56 | 45.35 | 26.16 | 9.67 |

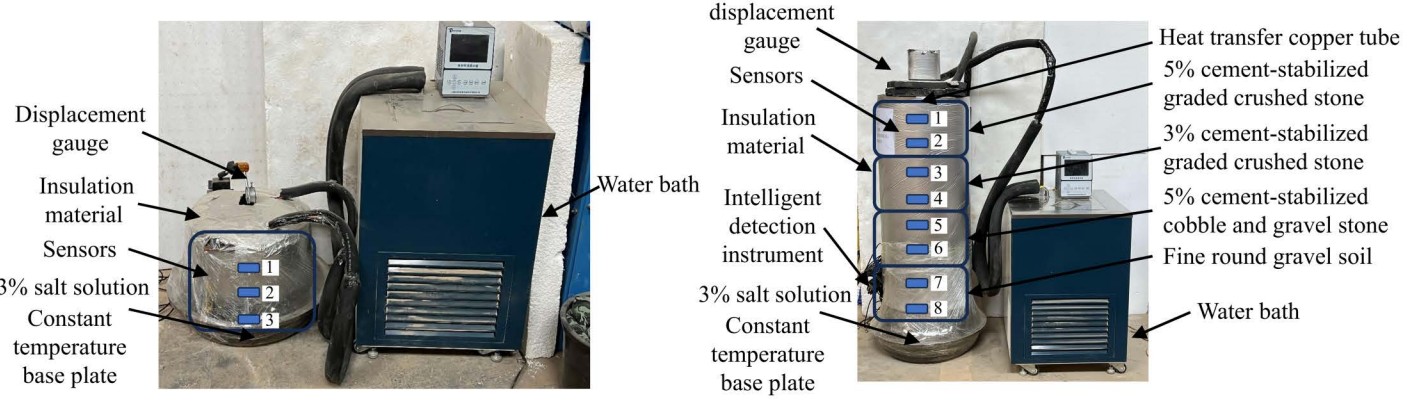

**Fig 1. Photos of freeze-thaw cycling test with soil column.**

content, and compaction degree are consistent with those of the single-layer soil column. During filling, sensors are buried at intervals of 15 cm starting 7.5 cm from the bottom of the mold, and a total of 8 sensors labeled from No. 1 to No. 8 from the top to bottom. The upper part of the sample is subjected to a temperature cycle of −30 °C to 40 °C, with the soil column model placed in a temperature-controlled room at 20 °C, while the lower part is maintained at room temperature. Each cycle consists of 96 hours, with 48 hours of freezing (−30 °C) followed by 48 hours of thawing (40 °C). The single-layer and multi-layer soil column freeze-thaw cycling tests are conducted for 3 and 7 freeze-thaw cycles, respectively.

The experimental setup includes temperature control equipment and data acquisition devices. Sensors are used to measure the moisture content, temperature, and electrical conductivity inside the filler material, while displacement gauges are employed to measure top deformation.

## 3. Experimental results and analysis

### 3.1 Temperature variation

The temperature-time curves of the single-layer and multi-layer soil columns are shown in Figs 2 and 3, respectively. The amplitude of temperature variations decreases upon increasing the distance from the top, and the difference in temperature extremes between adjacent sensors also decreases gradually. During freeze-thaw cycles, sensors No. 2 and below exhibit a temperature lag that became more significant as the distance from the top increases. The lag is also more pronounced when transitioning from hot cycles to cold cycles compared to the reverse transition. For the single-layer soil column, at a lower column height, the temperature variations of different fillers in each cycle are relatively small. For the multi-layer soil column, during the first cold cycle, the temperature of sensors No. 2 to No. 7 do not stabilize due to the initial temperature effect. Upon increasing the number of freeze-thaw cycles, the temperatures of the sensors during cold cycles continue to decrease and stabilize after the 4th cycle. The upper part of the soil column is the temperature-sensitive zone, while the lower part is the temperature-insensitive zone. In the temperature-sensitive zone, the soil is significantly influenced by the temperature cycles in the upper part, resulting in large temperature gradients.

### 3.2 Moisture content variation analysis

Fig 4 shows the variations in moisture content with height after three freeze-thaw cycles for different fillers, in which the moisture content decreases with the increase of the height for all fillers. For positions 2 and 3, the moisture content follows the trend: fine round gravel soil > 3% cement-stabilized graded crushed stone > 5% cement-stabilized graded crushed stone > 5% cement-stabilized cobble and gravel stone. The moisture content increased by 0.06–0.46%, 0.18–0.97%, 0.6–1.91%, and 2.39–4.17% for 5% cement-stabilized graded crushed stone, 3% cement-stabilized graded crushed stone, 5% cement-stabilized cobble and gravel stone, and fine round gravel soil, respectively. This indicates that the moisture content underwent a greater increase for loose materials than for stabilized solid fillers.

The moisture content variation with height after different freeze-thaw cycles for the multi-layer soil column is shown in Fig 5. Overall, the moisture content tends to increase upon increasing the number of freeze-thaw cycles, but the rate of increase slows upon increasing the number of cycles. After the freeze-thaw cycles, the moisture content at the top reaches 4.69%, while at the bottom, it is 7.95%. During the first and second cycles, sensors 3 and 5 registers a slight decrease in the moisture content due to cold-end aggregation and evaporation effects during the hot cycles at the top. This causes some moisture to migrate towards the top, but during subsequent freeze-thaw cycles, the moisture content at all heights increases. Sensors 1–5 show similar values, while sensors 6, 7, and 8 have larger differences. Sensor 8, being closest to the salt solution supply, exhibits the fastest increase in moisture content. The analysis investigates that during the first freeze-thaw cycle, the soil first undergoes a cooling phase, during which rapid freezing inhibits timely moisture migration.As time progresses and the temperature field stabilizes gradually, allowing pore water from the lower layers to migrate toward the cold end and accumulates at the freezing front (Fig 6). The internal water in the soil at the beginning of

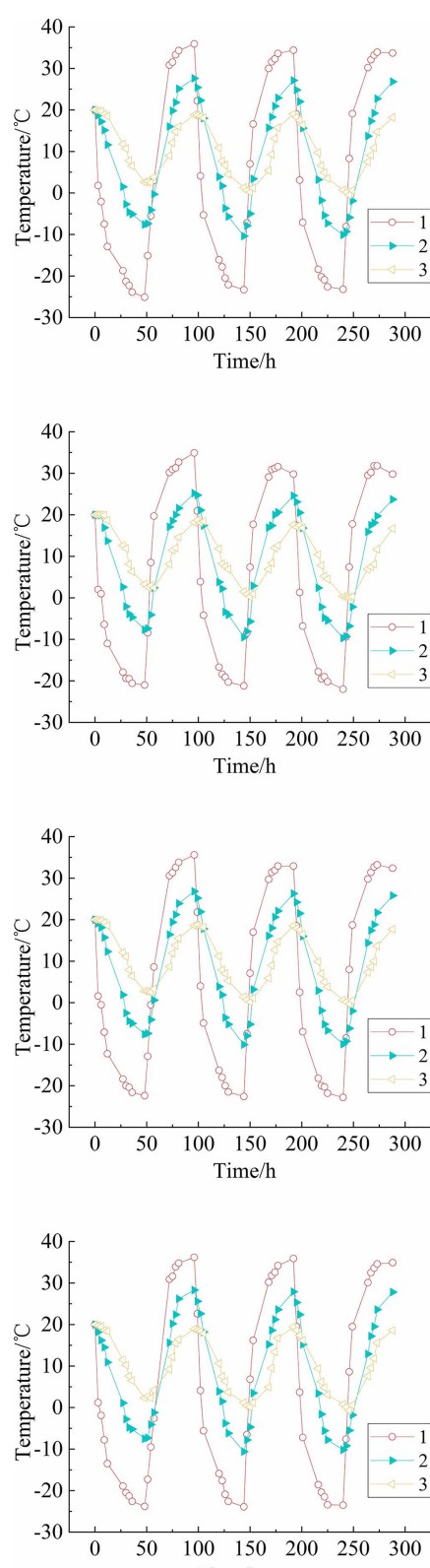

**Fig 2. Curves of soil column temperature versus time (single-layer).**

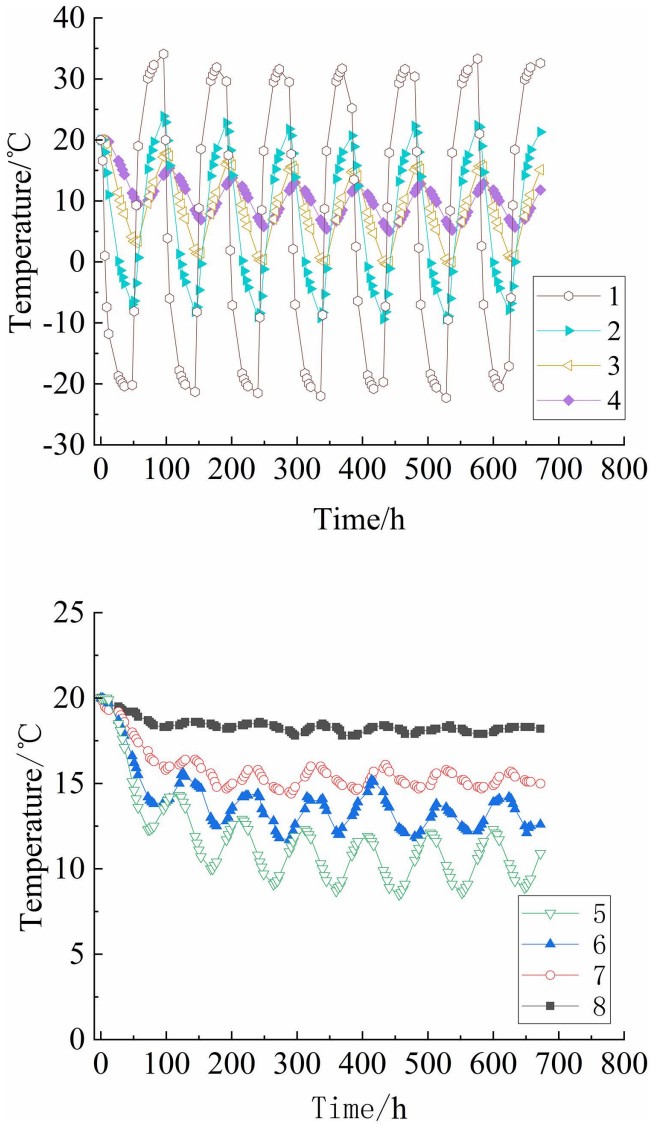

**Fig 3. Curves of soil column temperature versus time (multi-layer).**

the experiment is salt-free, but as the experiment progresses, salt begins to migrate with the water. As the salt content in the water increases with the number of freeze-thaw cycles, the freezing temperature gradually decreases, and the position of the freezing front shifts upward, reducing the distance from the cold end.

There is a higher moisture content at both ends and a lower content in the middle of the soil column after the third cycle. This difference increases with the number of freeze-thaw cycles due to the supply of salt solution at the bottom, which increases the moisture content. During the cooling phase of, this induces upward moisture migration towards the colder upper region., moisture consistently migrates from the warm end to the cold end. Specifically during cooling cycles, water from lower sections migrates upwards. After 48 hours of upward migration during the cooling phase, when the system transitions to the heating phase, the temperature in the upper soil column rises. During the cooling phase of cyclic freezing-thawing, moisture migrates towards the upper region. Throughout the freeze-thaw cycling process, moisture

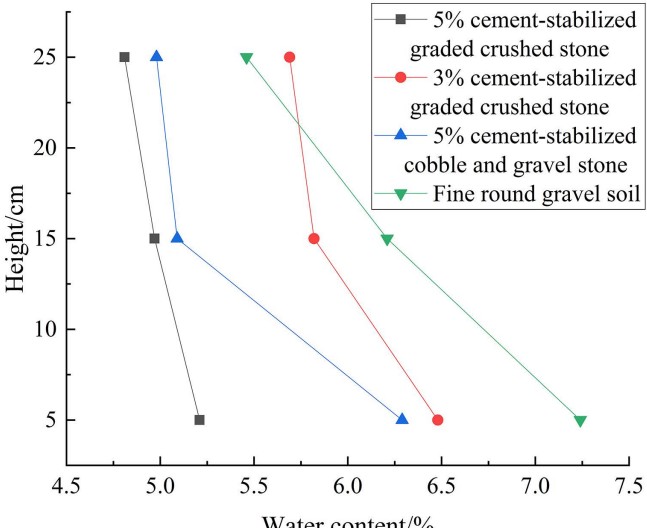

**Fig 4. Curves of water content of different fillers versus height after 3 freeze-thaw cycles (single-layer).**

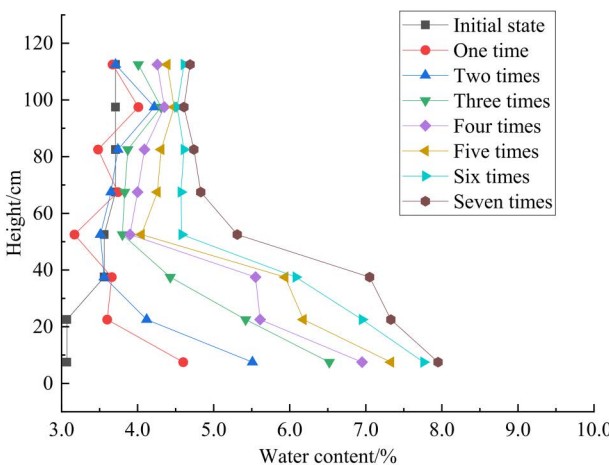

**Fig 5. Curves of water content versus height after different freeze-thaw cycles (multi-layer).**

migrates from the warm end to the cold end, and during the cold cycles, moisture from the bottom migrates upwards. After 48 hours of upward migration, the moisture enters the hot cycle as the temperature at the top of the soil column increases. This causes thawing of previously frozen soil, where ice crystals between soil particles melt into liquid water and subsequently percolate downward due to gravitational forces. However, the soil has insulating properties, and the melting of the soil in the upper part during the hot cycle gradually occures from the top to the bottom. When ice crystals in the upper soil melt and percolate to the frozen soil layer, ice crystals are formed by the water in the frozen soil and a temperature decrease causes salt crystals to precipitate. These crystals fill the gaps between soil particles, thereby blocking the water and salt migration pathways. As a result, some of the infiltrating liquid water is retained in this layer of soil.

After starting the freeze-thaw cycles in the soil column, the lower temperature-insensitive soil is less affected by temperature due to its distance from the cold and hot ends, resulting in the precipitation of fewer salt crystals caused

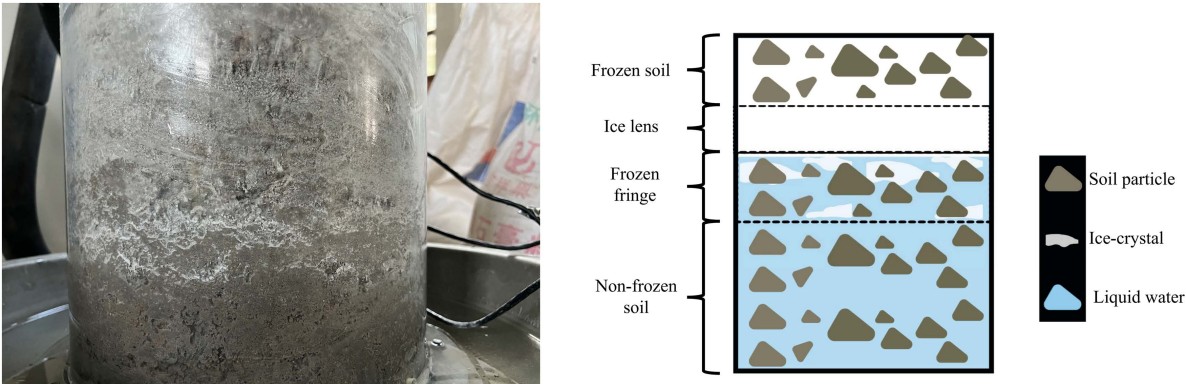

**Fig 6. Diagram of the freezing front.**

by water freezing and temperature-induced solubility changes. As the moisture content increases, the migration of externally-supplied solution into the soil column also weakens accordingly. At the end of the experiment, a significant lid effect is observed at the top of the soil column, where moisture in the soil is evaporated into the air. As the freeze-thaw system enters the cold cycle, the temperature at the top decreases, causing moisture in the air to condense on the surface of the soil column (Fig 7).

### 3.3 Salt content variation analysis

Fig 8 shows variations in the salt content versus height after 3 freeze-thaw cycles for different fillers. The vertical salt content follows a distribution similar to the increase in moisture content. For all fillers, there is a decrease in salt content with the height. Additionally, at the same position, the salt content follows the trend: fine round gravel soil > 5% cement-stabilized cobble and gravel stone > 3% cement-stabilized graded crushed stone > 5% cement-stabilized graded crushed stone. For the 5% cement-stabilized graded crushed stone, 3% cement-stabilized graded crushed stone, 5% cement-stabilized cobble and gravel stone, and fine round gravel soil, the salt content increase by 0.06–0.11%, 0.08–0.19%, 0.15–0.34%, and 0.21–0.49%, respectively.

Variations in the salt content versus height in a multi-layer soil column after freeze-thaw cycles are shown in Fig 9. After the freeze-thaw cycles, the salt content at the bottom of the soil column is 0.53%, while it is 0.03% at the top. The salt content in the upper region changes minimally, and upon increasing the number of freeze-thaw cycles, the upward migration of salt increases. During the fifth cycle, sensor 1 indicates a salt content change of 0.01%, indicating salt migration to the top. This suggests that during the freeze-thaw cycles, as moisture migrates upwards, salt also gradually migrates upwards. After the first freeze-thaw cycle, there is a significant change in salt content in the lower part and a minor change in the upper part due to the rapid freezing process where salt does not have enough time to migrate upwards. During the cold cycle, water above the freezing front formes ice crystals, causing salt to accumulate near the front and preventing diffusion into solid ice, resulting in a minor change in salt content in the upper part. The lower part has a supply of salt solution, leading to a greater change in the salt content due to the gradient effect and the continuous migration of water and salt. As the number of freeze-thaw cycles increases, the change in salt content decreases in the lower part, with significant changes mainly occurring in the middle of the soil column. This is because the continuous migration of water and salt intensifies, forming migration channels within the soil, and the melting of ice crystals in the upper part during the thaw cycle increases the moisture content and amount of dissolved salt. Some salt dissolves in water due to gravity and migrates downwards, resulting in a lower salt content in the upper part.

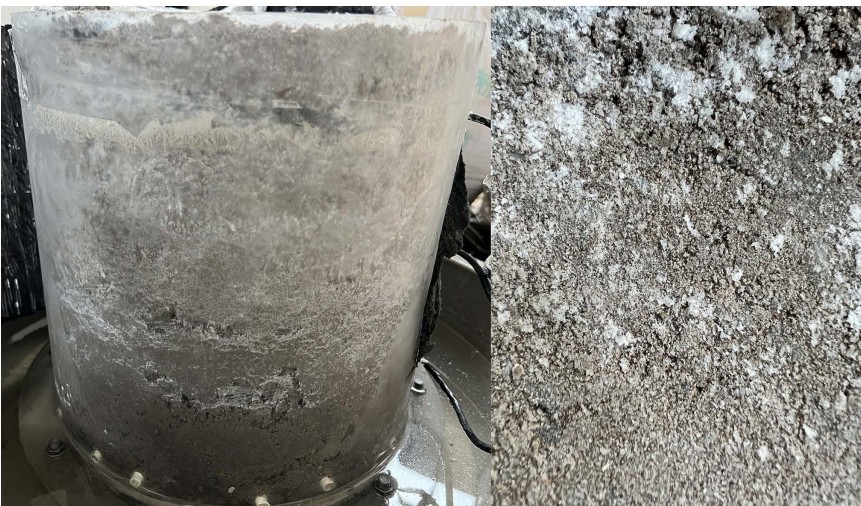

**Fig 7. Icing phenomenon on the top of the soil column.**

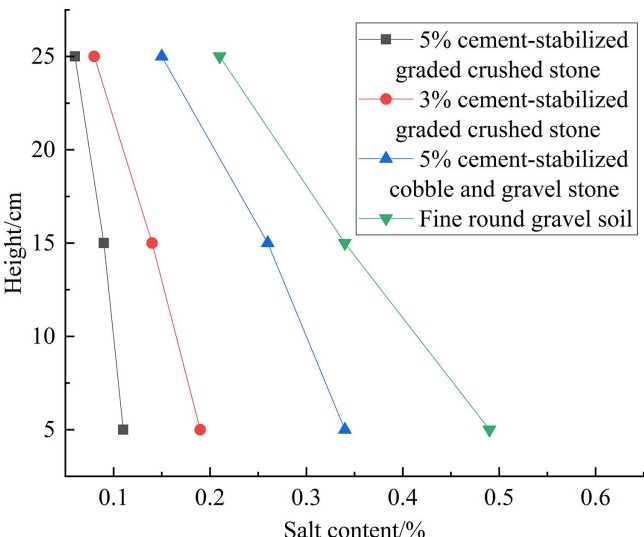

**Fig 8. Variation in salt content with height after 3 freeze-thaw cycles in different fillers (single-layer).**

After the experiment, some filler was dried using different drying methods: one-way drying and uniform drying. The salt analysis results of the dried samples are shown in Fig 10. During one-way drying with the heating source locateds at the right end, the water and salt inside the sample migrates in the opposite direction towards the heat source due to the effect of temperature. After the moisture has completely evaporated, salt particles precipitate on the sample surface. In contrast, in the case of uniform drying, the sample is heated evenly, causing moisture to evaporate uniformly from the sample surface and causing salt crystals to cover the surface evenly. This indicates that salt dissolves in water within the soil column filler and then migrates, eventually precipitating upon exceeding its saturation concentration.

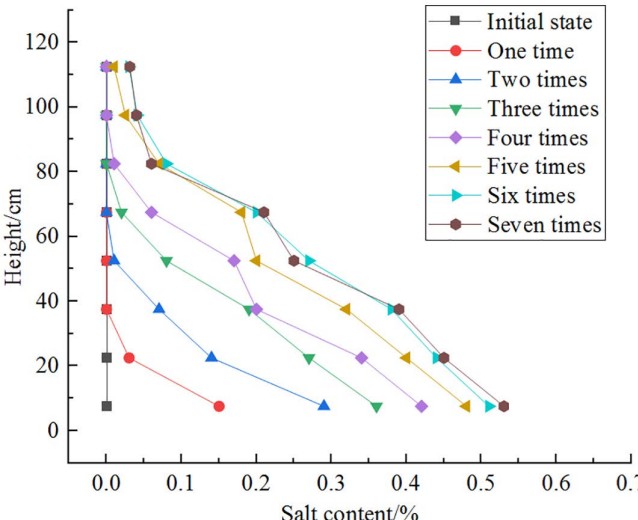

**Fig 9. Variation in salt content with height after different freeze-thaw cycles (multi-layer).**

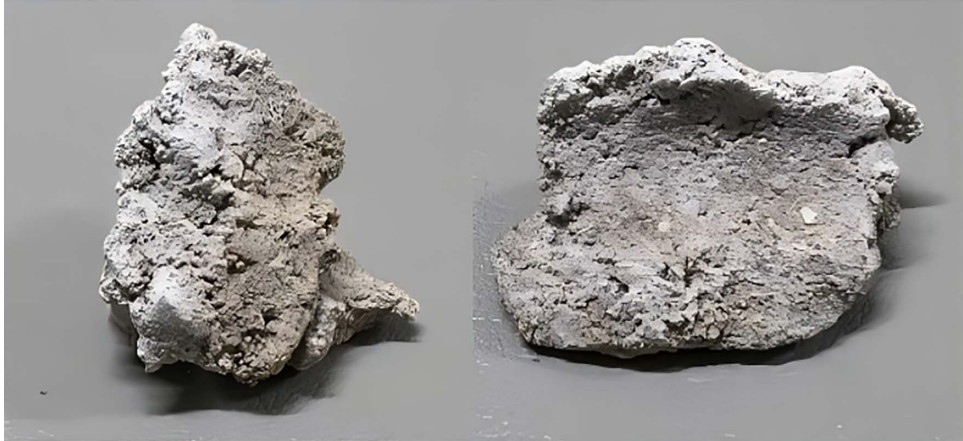

**Fig 10. Photos of salt precipitation on dried test blocks.**

### 3.4 Analysis of axial displacement changes

Fig 11 shows variations in the top displacement over time during freeze-thaw cycles for single-layer and multi-layer soil columns. Upon increasing the number of freeze-thaw cycles, the soil column initially settles and then undergoes increasingly severe cumulative residual deformation whose rate of increase gradually decreases. For the single-layer freeze-thaw cycle test with 5% cement-stabilized graded crushed stone, 3% cement-stabilized graded crushed stone, 5% cement-stabilized cobble and gravel stone, and fine round gravel soil, the cumulative residual deformations after 3 freeze-thaw cycles have shown in Table 2. This indicates that the solidified stable filler material exhibits lower residual deformation than the loose material. Excavation of the samples reveal that the solidified stable filler material remains intact without cracking. During the multi-layer freeze-thaw cycle test, the cumulative residual deformation after 7 cycles is 4.82 mm. The soil undergoes four main stages during freeze-thaw cycles: (1) a cooling and shrinkage stage, during which the soil

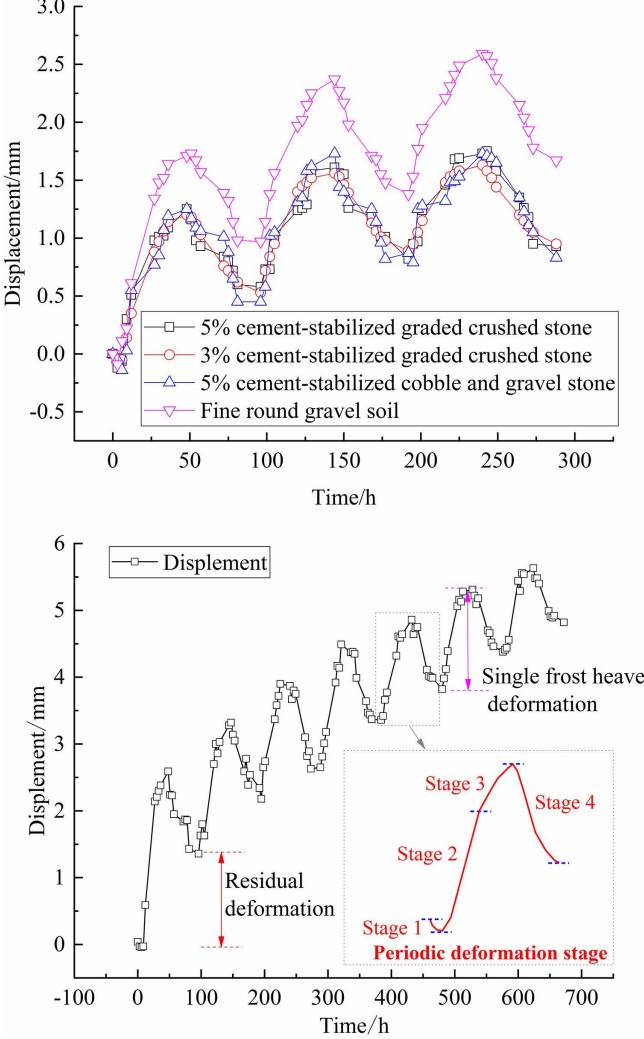

**Fig 11. Curves of top displacement with time.**

**Table 2. The cumulative residual deformations after 3 freeze-thaw cycles.**

| Filler Material | 5% cement-stabilized graded crushed stone | 3% cement-stabilized graded crushed stone, | 5% cement-stabilized cobble and gravel stone | fine round gravel soil |
|---|---|---|---|---|
| Cumulative residual deformation/mm | 0.93 | 0.95 | 0.83 | 1.67 |

temperature gradually decreases, causing a slight volume reduction due to cold shrinkage; (2) a salt expansion-erosion stage, during which salt crystals precipitate as the temperature decreases, leading to the formation of compounds such as ettringite due to reactions between sulfate ions and cement. This exerts crystallization pressure on surrounding particles; (3) a salt expansion-erosion-freezing expansion stage, during which the temperature reaches the freezing point of the salt solution, resulting in ice crystal formation, expansion, and compression of surrounding particles and causing structural damage; (4) a warming and melting stage, during which ice and salt crystals re-melt as the temperature increases, leading to liquid water formation, the gradual dissolution of salt crystals, and the release of crystallization pressure on the

soil particles, causing settling deformation. In engineering sites, the temperature difference can be controlled to reduce the deformation at each stage.

During the initial stages of freeze-thaw cycles, the precipitation of crystals from the solution provides support to the soil particles and counteracts the cold shrinkage effects, resulting in minimal initial settlement that is quickly offset by expansion deformation. However, during the warming phase, the internal temperature of the soil increases, leading to the re-melting of ice crystals and the dissolution of salt crystals, causing the soil to undergo melting settlement. The residual deformation after each freeze-thaw cycle accumulates due to the incomplete dissolution of salt crystals, and the presence of residual crystals provides support. This hinders the soil from fully recovering to its pre-freeze-thaw volume. With each subsequent cycle, the cumulative deformation increases, but the individual residual deformation decreases gradually.

## 4. Microanalysis

### 4.1 Analysis of biological components

After the experiment, a representative 3% cement-stabilized graded crushed stone layer is selected for X-ray diffraction analysis. The XRD pattern of the sample is shown in Fig 12, which indicates the simultaneous formation of ettringite and gypsum within the sample. The reaction between calcium ions in the cement hydration products and sulfate ions in the solution forms gypsum, which subsequently reacts with calcium hydroxide in the cement hydration products to form ettringite. When the concentration of sulfate ions in the solution is high, the chemical reaction to produce gypsum continues, resulting in residual gypsum during the formation of ettringite, which precipitates simultaneously with ettringite.

### 4.2 Microstructure analysis

Fig 13 shows SEM images of samples during freeze-thaw cycles. In the 500x magnified image in Fig 13(a), the soil particles are tightly connected, with a layer of cementitious material covering their surface, filling the gaps between soil particles. Additionally, there are pores of varying sizes between soil particles. During the cold cycle, salt precipitates, and upon increasing the temperature, the salt redissolves. During continuous freeze-thaw cycles, the salt and precipitates undergoes repeated precipitation-dissolution, which changes the soil structure. Soil particles experiences relative displacement,

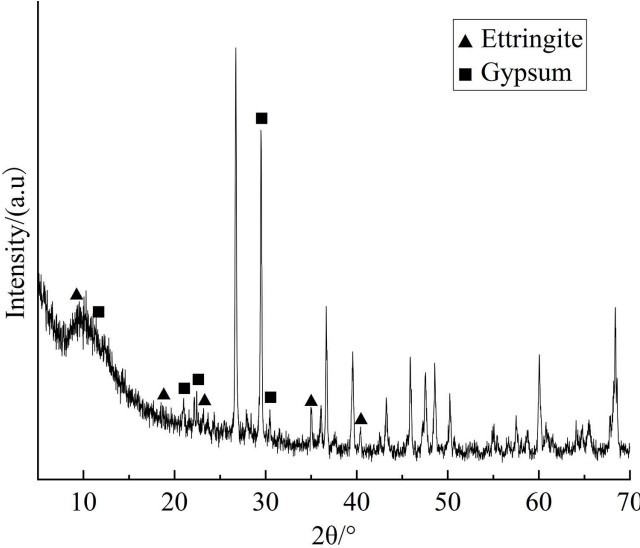

**Fig 12. XRD pattern of a representative 3% cement-stabilized graded crushed stone layer sample.**

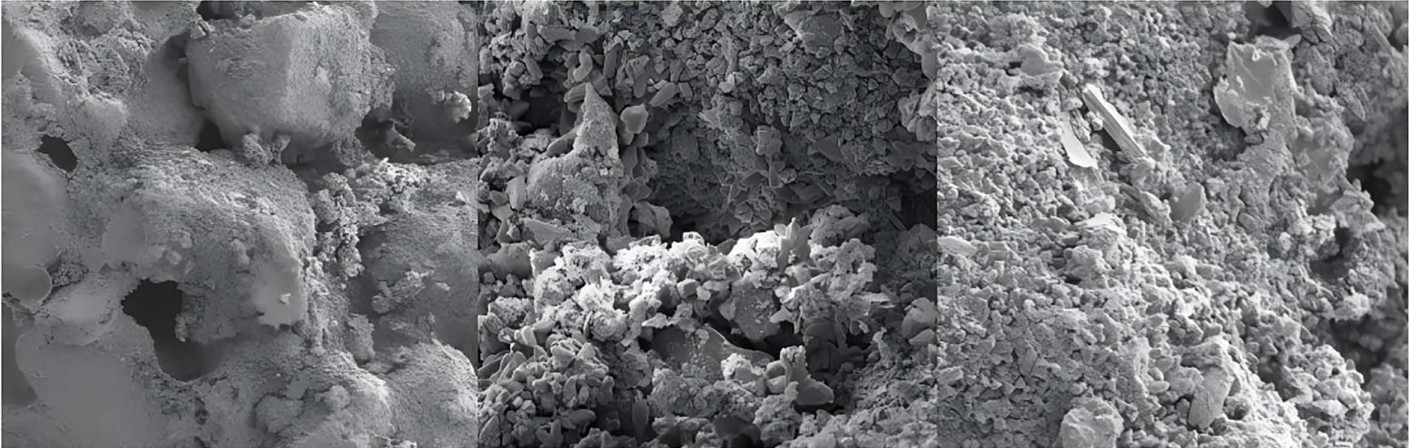

**Fig 13. SEM images of samples.**

and pores gradually form during water-salt migration, which serve as continuous migration channels. Upon further magnification, short columnar ettringite and gypsum crystals precipitate, which disrupt the connectivity between soil particles, leading to the formation of larger pores (Fig 13(b)). Flocculent precipitates appear between crystal particles, with a significant amount of salt precipitates adhering to the surface of soil particles and filling some of the pores (Fig 13(c)). This blocks some of the water-salt migration channels, thereby inhibiting water-salt migration.

## 5. Conclusion

This study investigated the arching disease on the transition section of a ballastless railway culvert in northwestern China by conducting freeze-thaw cycling tests on soil columns. The following conclusions are drawn:

In single-layer soil columns with different fillers, the moisture content decreases upon increasing the height. For multi-layer soil column, the moisture content generally increases with the freeze-thaw cycles, but the rate of increase slows as the number of cycles increase. The moisture content curve shows higher values at both ends and lower values in the middle. During warm cycles, waterlogging occurs inside the soil column, while freezing fronts and cap effects occurs during cold cycles.

In single-layer soil columns, the salt content decreases upon increasing the height for different fillers. For multi-layer soil column, salt migrates upwards during the freeze-thaw cycles, and the overall salt content increases continuously. During cold cycles, the freezing front also hinders salt migration to some extent. In both single-layer and multi-layer freeze-thaw cycles, the cumulative residual deformation increases with the number of cycles, but the rate of increase gradually slows. The freeze-thaw process mainly consists of a cooling and shrinkage stage, a salt expansion-erosion stage, a salt expansion-erosion-freezing expansion stage, and a warming and melting stage.

XRD analysis reveals the simultaneous generation of ettringite and gypsum within the samples. SEM analysis shows that the soil particles are covered with a layer of cementitious material that include some precipitated columnar ettringite and gypsum crystals due to water-temperature-salt migration.

In the soil column freeze-thaw cycling tests, only a single type of cement was used. Although the cement produced by different regions and manufacturers had the same grade, there were still some differences in their chemical compositions. The chemical reaction of sulfate erosion on cement-mixed graded crushed stones can still be analyzed qualitatively and quantitatively in detail.

On this basis, various prevention and control measures can be developed to prevent related diseases. It is worth further research to start from multiple aspects such as physics and chemistry.

## Supporting information

**S1 Data. Available data.**
(DOCX)

## Author contributions

**Conceptualization:** Wenhui Zhao, Ruiqi Wang.

**Investigation:** Hang Mu, Ke Zhang.

**Writing – original draft:** Wenhui Zhao, Ke Zhang.

**Writing – review & editing:** Hang Mu, Ruiqi Wang.

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
