## [Decision Letter · Decision Letter 0]

27 May 2025

Dear Dr. Zhang,

Thank you for submitting your manuscript to PLOS ONE. After careful consideration, we feel that it has merit but does not fully meet PLOS ONE’s publication criteria as it currently stands. Therefore, we invite you to submit a revised version of the manuscript that addresses the points raised during the review process.

We look forward to receiving your revised manuscript.

Kind regards,

Jiaolong Ren

Academic Editor

PLOS ONE

2. In the online submission form, you indicated that [Data will be made available on request].

Reviewers' comments:

Reviewer's Responses to Questions

**Comments to the Author**

1. Is the manuscript technically sound, and do the data support the conclusions?

Reviewer #1: Yes

Reviewer #2: Yes

2. Has the statistical analysis been performed appropriately and rigorously?

Reviewer #1: Yes

Reviewer #2: Yes

3. Have the authors made all data underlying the findings in their manuscript fully available?

Reviewer #1: Yes

Reviewer #2: Yes

4. Is the manuscript presented in an intelligible fashion and written in standard English?

Reviewer #1: Yes

Reviewer #2: Yes

Reviewer #1: Experimental study on water-heat-salt migration and deformation characteristics of subgrade filler during freeze-thaw cycles

Abstract

• The abstract provides a broad summary of the study, covering the motivation, methodology, and findings. However, the structure is imbalanced. The rationale (arching in culverts due to freeze-thaw cycles) is introduced well, but the objective is not clearly stated as a standalone sentence.

• The methods are implied but not concisely defined (e.g., freeze-thaw tests on soil columns).

• The findings are detailed, but too verbose for an abstract. Quantitative data is partially presented but would benefit from clearer summaries (e.g., numerical deformation values, migration rates).

• Clearly identify the research objective and condense findings into 2–3 data-backed sentences.

• Clearly articulate the objective of the study as a standalone sentence.

• Condense the description of findings; include only the most critical quantitative results.

• Maintain balance by briefly mentioning the methodology and limiting excessive detail in the findings.

Introduction

• The hypothesis and aim are implied but should be explicitly stated at the end of the section.

• Streamline the literature review by focusing on studies most relevant to multilayer subgrade behavior.

• Show the quantitative data of previous works and how they differ from your study.

• This study suggested to be used:

Saleh, S. A., Ismael, R. S. ., & Abas, B. S. . (2024). Effect of Soil Stabilization on Structural Design of Flexible Pavement. Journal of Studies in Civil Engineering, 1(1), 36–54. https://doi.org/10.53898/jsce2024113

Experimental Materials and Scheme

• Include a summary table detailing the freeze-thaw cycle parameters (temperature range, duration, number of cycles).

• Clarify the rationale for choosing specific cement stabilization percentages across materials.

• Mention whether replicate tests were conducted to ensure reproducibility.

Experimental Results and Analysis

Temperature Variation

• Support observed trends (e.g., temperature lag) with statistical validation or replicating data.

• Highlight how the findings align or contrast with existing models of thermal migration in soils.

Moisture Content

• Provide a comparative table summarizing moisture change across materials and cycle numbers.

• Consider discussing the implications of water redistribution on structural integrity more directly.

Salt Content

• Expand discussion on the practical impact of upward salt migration, especially in upper structural layers.

• Include statistical ranges or error bars in salt content measurements for clarity.

Deformation Analysis

• Present residual deformation results in tabular form for quick comparison across materials.

• Discuss how the four-stage deformation process informs design recommendations for embankments in similar regions.

Microanalysis

• Quantify microstructural features where possible (e.g., pore sizes, crystal dimensions).

• Consider correlating SEM/XRD findings more directly to macro-scale deformation behaviors.

• Specify the representativeness of the samples analyzed (e.g., selection criteria).

Conclusion

• Condense the list of findings and focus on the most impactful results.

• Add a final paragraph summarizing engineering implications and potential field applications.

• Briefly mention any limitations of the study and suggest future research directions, such as long-term field validation or model development.

Reviewer #2: Title and Abstract

The title is informative but could be more specific. For example, mention the geographic context (e.g., "in Northwestern China") or emphasize the sulfate erosion aspect more clearly.

The abstract is dense and includes too many technical details without sufficient explanation of their significance.

Can the abstract be revised to clearly define the research problem, methods, key findings, and implications in a more reader-friendly format?

What is meant by “arching disease” in the abstract? Consider defining this for readers unfamiliar with the term.

2. Introduction

The introduction provides a solid literature foundation but lacks a concise statement of the knowledge gap and hypothesis.

The "arching disease" is mentioned but not well contextualized.

Can the authors clearly articulate the specific research questions or hypotheses that this study seeks to answer?

What distinguishes this study from the numerous prior studies cited on water-salt migration and sulfate erosion?

Can the mechanism or definition of “arching disease” in transition sections be illustrated with a figure or schematic?

Experimental Materials and Scheme

The rationale for selecting the specific filler types and stabilization percentages is not explained.

Moisture contents appear unusually low (e.g., 3–5%). Clarify if this refers to dry densities or actual field conditions.

What criteria guided the selection of 3% and 5% cement contents? Are these field-validated values?

Why was 7 cycles chosen for the multi-layer test and 3 for the single-layer? Is there a theoretical or practical justification?

Is the 96-hour cycle duration realistic compared to field freeze-thaw rates?

Results and Analysis

Temperature Variation

Figures 2 and 3 are helpful but need better labeling and scale uniformity (legends? The numbering should be defined).

The description of "temperature lag" lacks quantification.

Can the authors quantify the temperature lag across sensors and compare it with model predictions or literature values?

Were thermal properties of the materials (e.g., thermal conductivity) considered in interpreting the lag?

Moisture Content Variation

The “high-low-high” water distribution is interesting but not fully explained.

The concept of "water lag" and "lid effect" is not defined clearly.

What mechanisms contribute to the “high-low-high” pattern: capillarity, thermal gradient, or osmotic flow?

How does the presence of salts affect freezing/melting behavior and water redistribution?

Salt Content Variation

Salt migration is well documented but would benefit from mass balance or modeling discussion.

The observed salt redistribution appears counterintuitive in later stages.

Can the authors provide a salt mass balance over cycles to show conservation or net accumulation?

How do salt precipitation/dissolution dynamics (e.g., solubility curves) explain the observed redistribution?

Deformation Analysis

The division into four freeze-thaw stages is insightful.

Deformation behavior is mainly described qualitatively.

Can the axial displacement be correlated with water/salt content or microstructural changes?

How repeatable or statistically significant are these deformation measurements?

Microanalysis

XRD and SEM results are interesting but poorly integrated with mechanical behavior analysis.

No discussion of quantitative phase analysis or crystal orientation/density.

What is the estimated amount of ettringite and gypsum formed, and how does this relate to observed swelling?

Is there any SEM or EDS evidence showing salt crystallization blocking pore throats?

Discussion (Missing/Underdeveloped)

There is no dedicated discussion section to synthesize findings or relate them to broader engineering practice.

What are the practical implications of these findings for high-speed railway subgrade design in saline soil regions?

How can the insights on moisture/salt migration inform predictive modeling or mitigation strategies?

Conclusion

The conclusions are mostly restatements of results.

Lacks discussion on limitations and future work.

Can the authors highlight the main engineering recommendations derived from this study?

What are the main limitations (e.g., lab scale vs. field scale, duration, boundary conditions)?

References and Style

The citation style is inconsistent (e.g., use of Chinese references with limited access).

Some citations appear repetitive or too many from the same group.

Can the authors diversify references with more international studies or review papers?

General/Editorial Suggestions

Improve language clarity: Terms like "occurres" and “slowes” suggest grammar and spell-checking is needed.

Add schematics of the experimental setup and sensor arrangement.

Ensure all figures are legible with consistent scales and labels.

**Do you want your identity to be public for this peer review?** For information about this choice, including consent withdrawal, please see our Privacy Policy

Reviewer #1: No

Reviewer #2: **Yes: ** Fidelis Odedishemi Ajibade

---

## [Author Response · Author response to Decision Letter 1]

10 Aug 2025

Dear editors,

Thank you for your valuable feedback and constructive suggestions on our manuscript, [Experimental study on water-heat-salt migration and deformation characteristics of subgrade filler during freeze-thaw cycles]. We sincerely appreciate the time and effort you and the reviewers have dedicated to evaluating our work.

In response to your comments, we have carefully revised the manuscript to address the points raised. Specifically, we have:

Thoroughly reorganized the content for improved clarity and flow.

Reformatted the manuscript to comply fully with the journal’s guidelines.

We believe these revisions have significantly enhanced the quality of the paper, and we are grateful for your insightful guidance. Please find our revised manuscript attached for your consideration. Should there be any further adjustments required, we would be happy to address them promptly.

Thank you once again for your support. We look forward to your feedback.

Best regard

Ke Zhang

---

## [Decision Letter · Decision Letter 1]

27 Aug 2025

Dear Dr. Zhang, 

We look forward to receiving your revised manuscript.

Kind regards,

Jiaolong Ren

Academic Editor

PLOS ONE

Journal Requirements:

Reviewers' comments:

Reviewer's Responses to Questions

**Comments to the Author**

Reviewer #1: All comments have been addressed

2. Is the manuscript technically sound, and do the data support the conclusions?

Reviewer #1: Yes

3. Has the statistical analysis been performed appropriately and rigorously?

Reviewer #1: N/A

4. Have the authors made all data underlying the findings in their manuscript fully available?

Reviewer #1: Yes

5. Is the manuscript presented in an intelligible fashion and written in standard English?

Reviewer #1: Yes

Reviewer #1: Thank you for revising the manuscript and for carefully responding to the reviewers’ comments. The revision is appreciated. However, I would like to suggest two points for further improvement:

1. In the response to each comment, it would be better to indicate the page and line numbers where the modifications have been made in the revised version. This will help the reviewers easily identify the changes and avoid any confusion.

2. The number of references should be increased where appropriate. Strengthening the literature support will improve the overall quality of the manuscript. I recommend adding the following references in the revised version:

o Saleh, S. A., Ismael, R. S., & Abas, B. S. (2024). Effect of Soil Stabilization on Structural Design of Flexible Pavement. Journal of Studies in Civil Engineering, 1(1), 36–54. https://doi.org/10.53898/jsce2024113

o Shirmohammadi, M., Hakimi Asiabar, S., and Peyrovi Cheshnasar, M. (2025). Dual Transitional Floors and Soil-Structure Synergy: A Paradigm Shift in Seismic Resilience for Hybrid High-Rise Systems. Advances in Civil Engineering and Environmental Science, 2(2), 62-71. doi: 10.22034/acees.2025.508498.1021

**Do you want your identity to be public for this peer review?** For information about this choice, including consent withdrawal, please see our Privacy Policy

Reviewer #1: No

---

## [Editor Report · Decision Letter 2]

5 Nov 2025

Experimental Study on Water-Heat-Salt Migration and Deformation Characteristics of Subgrade Filler during Freeze-Thaw Cycles in Northwestern China

PONE-D-25-25268R2

Dear Dr. Zhao,

We’re pleased to inform you that your manuscript has been judged scientifically suitable for publication and will be formally accepted for publication once it meets all outstanding technical requirements.

Kind regards,

Jiaolong Ren

Academic Editor

PLOS ONE
---

## [Editor Report · Acceptance letter]

PONE-D-25-25268R2

PLOS ONE

Dear Dr. Zhao,

I'm pleased to inform you that your manuscript has been deemed suitable for publication in PLOS ONE. Congratulations! Your manuscript is now being handed over to our production team.

Kind regards,

on behalf of

Dr. Jiaolong Ren

Academic Editor

PLOS ONE